# Enhanced Photocatalytic Activity and Stability of Bi_2_WO_6_ – TiO_2_-N Nanocomposites in the Oxidation of Volatile Pollutants

**DOI:** 10.3390/nano12030359

**Published:** 2022-01-23

**Authors:** Nikita Kovalevskiy, Svetlana Cherepanova, Evgeny Gerasimov, Mikhail Lyulyukin, Maria Solovyeva, Igor Prosvirin, Denis Kozlov, Dmitry Selishchev

**Affiliations:** 1Boreskov Institute of Catalysis, 630090 Novosibirsk, Russia; nikita@catalysis.ru (N.K.); svch@catalysis.ru (S.C.); gerasimov@catalysis.ru (E.G.); lyulyukin@catalysis.ru (M.L.); smi@catalysis.ru (M.S.); prosvirin@catalysis.ru (I.P.); kdv@catalysis.ru (D.K.); 2Research and Educational Center “Institute of Chemical Technologies”, Novosibirsk State University, 630090 Novosibirsk, Russia

**Keywords:** photocatalysis, VOC oxidation, UV, visible light, composite photocatalyst, N-doped TiO_2_, Bi_2_WO_6_, stability test

## Abstract

The development of active and stable photocatalysts for the degradation of volatile organic compounds under visible light is important for efficient light utilization and environmental protection. Titanium dioxide doped with nitrogen is known to have a high activity but it exhibits a relatively low stability due to a gradual degradation of nitrogen species under highly powerful radiation. In this paper, we show that the combination of N-doped TiO_2_ with bismuth tungstate prevents its degradation during the photocatalytic process and results in a very stable composite photocatalyst. The synthesis of Bi_2_WO_6_–TiO_2_-N composites is preformed through the hydrothermal treatment of an aqueous medium containing nanocrystalline N-doped TiO_2_, as well as bismuth (III) nitrate and sodium tungstate followed by drying in air. The effect of the molar ratio between the components on their characteristics and photocatalytic activity is discussed. In addition to an enhanced stability, the composite photocatalysts with a low content of Bi_2_WO_6_ also exhibit an enhanced activity that is substantially higher than the activity of individual TiO_2_-N and Bi_2_WO_6_ materials. Thus, the Bi_2_WO_6_–TiO_2_-N composite has the potential as an active and stable photocatalyst for efficient purification of air.

## 1. Introduction

Photocatalytic oxidation is a convenient method for removing volatile contaminants from air via their complete degradation. TiO_2_ is the most used photocatalyst for the oxidative degradation of pollutants due to its high activity, stability, and safety for living organisms [1,2]. A high activity of TiO_2_ results from the positions of its valence band (VB) and conduction band (CB) that are appropriate for the generation of charge carriers with high redox potentials, an extended surface area, and surface chemistry [3,4]. The potentials of photogenerated electrons and holes are high enough for the formation of reactive oxygen species, which can provide oxidative degradation of various types of pollutants [5,6]. The degradation of aromatics, ketones, aldehydes, alcohols, carboxylic acids, alkanes, alkenes, and heterocompounds was investigated and discussed in the previously published papers [7,8,9,10,11,12,13,14]. However, a wide band gap of TiO_2_ (3.0–3.2 eV) does not allow for efficient utilization of sunlight because TiO_2_ can absorb radiation with wavelengths shorter than 400 nm (i.e., UV light) that corresponds to less 5% of the total solar light [15]. Room light sources also have a low impact on TiO_2_ because they emit photons in the visible region of spectrum. Therefore, the development of active and stable photocatalysts, which can efficiently utilize light in a wide range, is important for environmental protection and human safety.

One of the promising photocatalysts for pollutant degradation both under UV and visible light is TiO_2_ doped with nitrogen [16,17,18,19]. Energy levels of nitrogen impurities are located in the band gap of TiO_2_ above its VB and lead to a reduction in the energy required for the excitation of electrons [20]. The potential of photogenerated holes localized in nitrogen levels is still high enough for the direct oxidation of organic molecules adsorbed on the surface of the photocatalyst [21]. As a result, N-doped TiO_2_ is able to completely oxidize organic pollutants under radiation with wavelengths up to 500–530 nm and exhibits high values of quantum efficiency [22]. N-doped TiO_2_ can be prepared using inexpensive inorganic precursors through simple routes that make it significant for practical application [23].

However, the efficiency of light utilization by N-doped TiO_2_ in the visible region of the spectrum is still far from the corresponding value for the utilization of the UV light attributed to the band-to-band excitation of electrons in TiO_2_ [19,24]. An increase in the values of quantum efficiency in the visible region is required for better utilization of solar light by this type of photocatalysts. It stimulates further studies on the modification of N-doped TiO_2_ to enhance its photocatalytic activity. Another aspect of N-doped TiO_2_ is that a gradual decrease in its visible-light activity occurs under long-term irradiation. This deactivation results from the oxidation of nitrogen species in the TiO_2_ lattice by the photogenerated holes that do not migrate to the surface of the photocatalyst and do not participate in the redox reactions with adsorbed molecules [25]. Fast transfer of holes from the TiO_2_ phase may help to prevent the degradation of nitrogen species and deactivation of the photocatalyst.

The surface modification of N-doped TiO_2_ may solve both the mentioned problems via an enhanced interface transfer of photogenerated charge carriers that would improve their separation and prevent fast recombination. Surface modification with transition metals, especially noble metals, is an efficient way to suppress the recombination of charge carriers [26]. However, this approach commonly affects the pathway of photogenerated electrons only because an interface transfer of electrons from the CB of TiO_2_ to the metal species occurs in the metal-modified photocatalyst. As a result, the modification with metals leads to an increase in the efficiency of light utilization but it has a low effect on the stability of N-doped TiO_2_ [27].

Another approach to enhance the separation of charge carries is the combination of TiO_2_ with other semiconductors. When two semiconductors are in contact, different types of heterojunctions can occur [28,29,30]. The type of heterojunction depends on the characteristics of semiconductors that include their band gaps, potentials of VB and CB, and work functions [31,32]. A good cocatalyst for N-doped TiO_2_ would be a semiconductor with a band gap that is narrower than the band gap of TiO_2_, while the position of its VB would be close to the VB of TiO_2_. These parameters would give a possibility to obtain a composite material, which is able to generate holes with a potential high enough for the formation of OH^•^ species and direct oxidation of organic compounds [4].

Bismuth and tungsten oxides with a band gap of 2.6–2.8 eV correspond to the mentioned requirements [33]. Both type II [34] and S-scheme [35] heterojunctions were proposed for the composite Bi_2_O_3_–TiO_2_ system. Similarly, a heterojunction of type II [36] or Z-scheme [37] was proposed for the WO_3_–TiO_2_ composite. An increased photocatalytic activity in all cases was attributed to an enhanced separation of charge carriers due to interface transfer.

Bismuth tungstate (Bi_2_WO_6_) is also regarded as one of the promising semiconductors which can enhance the characteristics of TiO_2_ and N-doped TiO_2_ on the efficiency of light utilization. Bi_2_WO_6_ absorbs light with wavelengths up to 450 nm and has a photocatalytic activity in the oxidation of organic pollutants both under UV and visible light [38]. On the other hand, the potential of photogenerated electrons in Bi_2_WO_6_ is not high enough for efficient transfer to oxygen molecules if compared with TiO_2_-based photocatalysts. As a result, pristine Bi_2_WO_6_ does not exhibit superior values of photocatalytic activity in the degradation of pollutants. The composite Bi_2_WO_6_–TiO_2_ photocatalysts was previously shown to exhibit substantially higher activity than parent semiconductors [39,40,41,42]. In contrast to photocatalytic activity, no information concerning the effect of Bi_2_WO_6_ on the stability of N-doped TiO_2_ was found by the authors during the literature review. It was our motivation to perform this study and investigate the stability of Bi_2_WO_6_–TiO_2_-N composites under long-term exposure to powerful radiation.

In this paper, we show the effect of the ratio between Bi_2_WO_6_ and N-doped TiO_2_ components on the activity of the composite system and its stability during long-term experiments. The synthesis of bismuth tungstate and composites with N-doped TiO_2_ was performed using the hydrothermal method. The experimental data shows that the combination of N-doped TiO_2_ with bismuth tungstate via this method results in a stable composite photocatalyst due to the prevention in the degradation of nitrogen species in TiO_2_ during the photocatalytic process. In addition to an enhanced stability, the composite photocatalysts with a low content of Bi_2_WO_6_ also exhibit an enhanced activity that is substantially higher than the activity of the pristine N-doped TiO_2_ and Bi_2_WO_6_ photocatalysts. 

## 2. Materials and Methods

### 2.1. Synthesis of Photocatalysts

High purity grade titanium(IV) oxysulfate (TiOSO_4_), bismuth(III) nitrate pentahydrate (Bi(NO_3_)_3_·5H_2_O), and sodium tungstate dihydrate (Na_2_WO_4_·2H_2_O), purchased from Sigma-Aldrich (Saint Louis, MO, USA), as well as reagent grade ammonium hydroxide solution (NH_4_OH, 25%) and nitric acid (HNO_3_, 65%) purchased from AO Reachem Inc. (Moscow, Russia), were used for the synthesis as received without further purification.

N-doped TiO_2_ (TiO_2_-N) was prepared from an aqueous solution of TiOSO_4_ via the precipitation using ammonia according to our previously published technique [22]. Briefly, the solutions of titanium oxysulfate (1 M, 150 mL) and ammonium hydroxide (4 M) were simultaneously added dropwise to deionized water (200 mL) under vigorous stirring. The pH during the precipitation was adjusted at 7 by tuning the flows of reagents. After storing for 48 h in mother liquor, the precipitate was separated by centrifugation and washed with deionized water. Finally, it was calcined in air at 450 °C for 3 h and grinded using an agate mortar. Large particles were separated using a sieve to get a fraction of fine particles.

The synthesis of Bi_2_WO_6_ and its composites with N-doped TiO_2_ was performed via the precipitation from an acidic solution of Bi(NO_3_)_3_ using Na_2_WO_4_ followed by the hydrothermal treatment. The molar ratio between the components was varied in a wide range. Table 1 shows the mass values of all the precursors used during the synthesis of samples. Bismuth nitrate (m(X) g) was dissolved in nitric acid solution (1 M, 40 mL) and prepared TiO_2_-N (m(Z) g) was suspended in this solution. Then, a solution of Na_2_WO_4_ (m(Y) g in 10 mL deionized water) was added dropwise to the prepared suspension under continuous stirring. The suspension was transferred to a Teflon-lined autoclave and thermally treated at 160 °C for 10 h. After hydrothermal treatment, the precipitate was separated by centrifugation and washed with deionized water followed by final drying and grinding. Actual Bi_2_WO_6_ contents estimated using X-ray fluorescence analysis corresponded to the loading of components. The Bi_2_WO_6_ photocatalyst was synthesized similarly without the addition of TiO_2_-N. 

### 2.2. Characterization Techniques

Powder X-ray diffraction (XRD) was used for analysis of crystal phases in the prepared photocatalysts. The data were collected using a D8 Advance diffractometer (Bruker, Billerica, MA, USA), which was equipped with a CuK_α_ radiation source and a LynxEye position sensitive detector, in the 2*θ* range of 10–75° with a step of 0.05° and a collection time of 3 s. The average size of TiO_2_ crystallites was estimated from the collected XRD patterns using TOPAS software (Bruker, Billerica, MA, USA). The specific surface area of the samples was determined by BET analysis of N_2_ isotherms measured at 77 K using an Autosorb-6B-Kr (Quantachrome Instruments, Boynton Beach, FL USA). Before the measurements, the samples were outgassed in a vacuum at 150 °C for 17–23 h. The pore volume in the samples was estimated from the isotherms using the BJH method as a cumulative desorption pore volume. The morphology of individual components and composite photocatalysts was investigated using scanning electron microscopy (SEM). SEM micrographs were received using an ultra-high resolution Field-Emission SEM (FE-SEM) Regulus 8230 (Hitachi, Tokyo, Japan) at an accelerating voltage of 5 kV. The local structure was investigated by transmission electron microscopy (TEM) using a JEOL-2010 microscope (JEOL, Tokyo, Japan) at 200 kV and a resolution of 0.14 nm. The surface composition was investigated by X-ray photoelectron spectroscopy (XPS) using a SPECS photoelectron spectrometer (SPECS Surface Nano Analysis GmbH, Berlin, Germany) equipped with a PHOIBOS-150 hemi-spherical energy analyzer and an AlKα radiation source (hν = 1486.6 eV, 150 W). The scale of binding energy (BE) was pre-calibrated using the photoelectron lines of Au4f7/2 (84.0 eV) and Cu2p3/2 (932.67 eV) from metallic gold and copper foils. Peak fitting in the collected spectra was performed using XPSPeak 4.1 software (Informer Technologies Inc., Los Angeles, CA, USA). The optical properties of the samples were analyzed at room temperature using UV-Vis diffuse reflectance spectroscopy (DRS). The spectra were recorded in the range of 250–850 nm on a Cary 300 UV-Vis spectrophotometer from Agilent Technologies Inc. (Santa Clara, CA, USA) equipped with a DRA-30I diffuse reflectance accessory and special pre-packed polytetrafluoroethylene (PTFE) as a reflectance standard. The optical band gap was estimated using the Tauc method based on an assumption of indirect allowed excitations.

### 2.3. Photocatalytic Experiments

Long-term photocatalytic experiments were performed in a continuous-flow setup using acetone as a test volatile organic compound. The schematic diagram of this setup and details on the operation process can be found in Appendix A. Validation of the setup was previously performed during the investigation of functional properties of various photocatalytic systems [43,44,45,46]. The rate of air flow was 67 ± 1 mL min^−1^ and the inlet concentration of acetone vapor was ca. 32 µmol L^−1^. The relative humidity during experiments was adjusted at 19 ± 1% by tunning the flows of dry and humidified air. The experiments were performed with a thick layer of photocatalyst at a geometric area of 9.1 cm^2^ and its area density was 20 mg cm^−2^ to absorb the maximum number of photons. A UV light-emitting diode (LED) with a maximum at 371 nm and a blue LED with a maximum at 450 nm (see Appendix A) were used for the irradiation of photocatalysts and for the evaluation of their activity in both UV and visible regions, respectively. Specific radiation powers measured using an ILT950 spectroradiometer (International Light Technology, Peabody, MA, USA) were 10 and 160 mW cm^−2^ for UV and blue lights, respectively. Analysis of acetone and oxidation products in the gas phase was performed using in situ IR spectroscopy on an FTIR spectrometer FT-801 (Simex LLC, Novosibirsk, Russia) equipped with an IR long-path gas cell (Infrared Analysis Inc., Anaheim, CA, USA). The concentrations of volatile compounds were determined from the collected IR spectra using the Beer-Lambert law. The photocatalytic activity of materials was estimated as the steady-state rate of CO_2_ formation (μmol min^−1^), which was the final oxidation product. Based on the statistics of many experiments, a relative error estimated using the confidence interval for the probability of 95% during the measurement of photocatalytic activity in the setup did not exceed 5%. This error value was used for all tested samples to evaluate statistically significant differences in their activity.

## 3. Results and Discussion

Composite photocatalysts containing Bi_2_WO_6_ were synthesized using a wet process with the purpose to enhance the activity and stability of N-doped TiO_2_ as a photocatalyst for the degradation of volatile organic pollutants under visible light. Correlations between physicochemical characteristics of the materials and their functional properties are discussed below.

### 3.1. Characteristics

Phase composition, morphology, and textural and optical properties commonly have a strong effect on the photocatalytic ability of semiconducting materials. Therefore, these properties were investigated for the synthesized samples using corresponding techniques.

Figure 1a shows the XRD patterns of single photocatalysts and certain samples from a series of synthetized composites. TiO_2_-N prepared from an aqueous solution of titanium oxysulfate had the crystal structure of anatase because all peaks in its XRD pattern are coincided with the reflexes attributed to anatase. Main reflection peaks at 2Θ of 25.3, 37.8, 48.0, 53.9, and 55.1° correspond to the (101), (004), (200), (105), and (211) diffraction planes of anatase TiO_2_, respectively. No peaks attributed to the rutile phase were detected. Formation of anatase phase at relatively low temperatures is typical for the preparation of TiO_2_ from the titanium oxysulfate precursor [47,48]. An averaged size of anatase crystallites in the TiO_2_-N sample was estimated as 20 nm. SEM micrographs in Figure 2a illustrate this fact and show that large agglomerates in TiO_2_-N consist of adherent nanosized spherical and oval particles, wherein the space between forms a porous structure. Consequently, this sample had a high surface area (98 m^2^ g^−1^) and pore volume (0.54 cm^3^ g^−1^) that positively affected its photocatalytic ability.

As expected, the applied technique via hydrothermal treatment of aqueous solutions of bismuth nitrate and sodium tungstate resulted successfully in a formation of orthorhombic bismuth tungstate (Bi_2_WO_6_, PDF#39-0256). The main reflection peak at 2Θ of 28.3° corresponds to the (131) plane of orthorhombic Bi_2_WO_6_, while the peaks with maxima at 32.9 and 47.2° result from a series of reflexes: {(200), (002), (060)} and {(202), (260)}, respectively. No marked peaks attributed to the individual bismuth and tungsten oxides were detected. On the other hand, an asymmetry of peaks attributed to multiplied reflexes of Bi_2_WO_6_ was observed in the XRD pattern of the synthesized Bi_2_WO_6_ sample (Figure 1b). This fact indicates an anisotropy in the shape of its particles (see Appendix A for details). The results of microscopic study using the SEM technique confirms this statement. Figure 2b shows that the Bi_2_WO_6_ sample has a lamellar structure and its agglomerates consist of adherent nanoplates, which have a thickness lower than 20 nm (see the bottom micrograph received at a high magnification). To illustrate the effect of anisotropic sizes of crystallites on XRD patterns, Figure 1c shows the XRD pattern simulated for the nano-crystallites of Bi_2_WO_6_, which have the shape of disks (see Appendix A for details). A good accordance with the experimental pattern was observed for the model corresponding to a small thickness of disks (6.6 nm) but a large diameter (30 nm). These values characterize the relative sizes of crystallites in the Bi_2_WO_6_ sample at different crystallographic axes. Saha et al. [49] have recently shown using in situ X-ray techniques the mechanism of nucleation and growth of Bi_2_WO_6_ crystallites, describing the predominant formation of nanoplatelets.

In the case of Bi_2_WO_6_–TiO_2_-N composites, the phase composition was influenced by the content of Bi_2_WO_6_. Only the peaks attributed to the anatase phase were observed in the XRD pattern of the sample with a Bi_2_WO_6_:TiO_2_-N molar ratio of 1:100 and no additional peaks were detected. Additionally, the volume of the TiO_2_ unit cell did not change compared to pristine TiO_2_-N, which indicates no doping TiO_2_ lattice with Bi or W-species. This means that at low amounts of Bi and W-precursors, the formation of Bi_2_WO_6_ crystallites in a high number does not occur and they seem to be in the form of clusters on the extended surface of TiO_2_-N particles.

An increase in the molar ratio up to 5:100 resulted in the appearance of the orthorhombic phase of Bi_2_WO_6_ in the composition (Figure 1a). This means that the amounts of Bi and W-precursors in this case are high enough for the formation of Bi_2_WO_6_ crystallites in a high number. Figure 2c shows a hybrid structure of the Bi_2_WO_6_–TiO_2_-N (5:100) sample, where small TiO_2_ nanoparticles are located on the surface of Bi_2_WO_6_ nanosheets of big size, and they together form large porous agglomerates. In addition, Figure 3 shows a local structure of the Bi_2_WO_6_–TiO_2_-N (5:100) composite studied using the TEM technique. A high number of nano-crystallites with a lattice spacing of 0.35 nm attributed to the (101) plane of anatase TiO_2_ was observed on the surface of sheet-like particles of Bi_2_WO_6_.

The results of XPS analysis also confirm the presence of bismuth and tungstate elements on the surface of the Bi_2_WO_6_–TiO_2_-N (5:100) sample. According to the peak positions (Figure 4a,b), they have high charge states (i.e., +3 for Bi and +6 for W), which correspond to the states in the chemical composition of Bi_2_WO_6_. No additional lines attributed to the other charge states of these elements were detected in corresponding spectral regions. These data support the results of other characterization techniques on the formation of the Bi_2_WO_6_ semiconductor, which forms a composite system with TiO_2_-N. The structure of the synthesized Bi_2_WO_6_–TiO_2_-N composite indicates a possibility for a heterojunction of photogenerated charge carriers between the Bi_2_WO_6_ and TiO_2_-N semiconducting components under irradiation and for an enhancement in its photocatalytic ability, as will be discussed later.

All samples with a molar ratio of higher than 5:100 had the crystal phases of both components (i.e., orthorhombic bismuth tungstate and anatase). An increase in the content of Bi_2_WO_6_ led to an increase in the fraction of the Bi_2_WO_6_ crystal phase and, consequently, to a decrease in the textural characteristics of composites because synthesized Bi_2_WO_6_ had substantially worse textural properties than TiO_2_-N due to a lamellar structure. As mentioned above, the synthesized TiO_2_-N sample had a high surface area (98 m^2^ g^−1^) and pore volume (0.54 cm^3^ g^−1^) due to an extended pore structure formed between adherent nanosized particles of TiO_2_. The specific surface area and pore volume of Bi_2_WO_6_ were 34 m^2^ g^−1^ and 0.20 cm^3^ g^−1^, respectively, which corresponds to high values for the concerned material [41,42]. As the molar ratio between Bi_2_WO_6_ and TiO_2_-N was increased, the textural characteristics of composites monotonically decreased down to the values attributed to pristine Bi_2_WO_6_ because the content of the superfine component was decreased (Figure 5). Commonly, photocatalysts with an extended surface area exhibit higher photocatalytic activity due to the higher number of sites for adsorption of oxidizing compounds. Therefore, the Bi_2_WO_6_–TiO_2_-N composites with a low content of Bi_2_WO_6_ would be preferable in this aspect.

It is important to note that in addition to the main elements (Bi, W, Ti, and O) in the chemical composition of Bi_2_WO_6_–TiO_2_-N (5:100), XPS analysis of this sample shows a signal in the N1s spectral region at 399.8 eV attributed to nitrogen species (Figure 4c). We have previously shown [22] that this value of binding energy corresponds to nitrogen that is placed in an interstitial position in the TiO_2_ lattice and has a weak positive charge due to interactions with lattice O atoms. The formed nitrogen species result in the appearance of additional energy levels (π* N-O) in the band gap of TiO_2_, above its VB [50], and provide the absorption of light in the visible region of the spectrum due to the excitation of electrons from these levels to the CB of TiO_2_. In the UV-vis spectrum of synthesized TiO_2_-N (Figure 6a), it appears as a shoulder of absorption in the region of 390–530 nm in addition to fundamental (i.e., band-to-band) absorption in TiO_2_-N. Approximation of the Tauc plot for this sample (Figure 6b) gives a value of 3.18 eV, attributed to the band gap of anatase, but light absorption also occurs in the low-energy region until 2.32 eV (i.e., 535 nm). This value corresponds to a minimum energy required for the photoexcitation of electrons in N-doped TiO_2_ and can be used for an estimation of the position of nitrogen energy levels compared to the bands of TiO_2_.

Bi_2_WO_6_ has a narrower band gap than TiO_2_ and can absorb the light of the visible region up to 450 nm due to the band-to-band excitation of electrons. An estimated value of its optical band gap was 2.80 eV (Figure 6b). Considering the ability of single materials to absorb the light of the blue region, Figure 6a shows that TiO_2_-N can absorb a much higher number of photons in this region than Bi_2_WO_6_. This can be a reason for the higher rate in the photocatalytic oxidation of acetone vapor over TiO_2_-N compared to the Bi_2_WO_6_ photocatalyst, as will be discussed in the next section.

In contrast to pristine Bi_2_WO_6_, UV-vis spectra of composite Bi_2_WO_6_–TiO_2_-N photocatalysts had a form that is similar to the spectrum of TiO_2_-N (Figure 6a). The samples with a low content of Bi_2_WO_6_ (i.e., molar ratio of 1:100 and 5:100) exhibited even higher absorption of blue light due to a combination with Bi_2_WO_6_ and a greater depth for the penetration of light. Further increase in the content of Bi_2_WO_6_ led to lower values of absorption due to a decrease in the content of TiO_2_-N, which better absorbed light in this region. However, for all these composite photocatalysts, the minimum energy required for the photoexcitation of electrons was ca. 2.3 eV, which is similar to TiO_2_-N. This indicates that the proposed Bi_2_WO_6_–TiO_2_-N composite system has the potential to perform the photocatalytic reactions under solar radiation, which has the major content in the visible region.

All synthesized photocatalysts absorb light both in UV and blue regions but different pathways of excitation are realized. Therefore, their photocatalytic activity was investigated independently in each region to correctly analyze the effect of composition on the functional properties of materials.

### 3.2. Photocatalytic Activity

TiO_2_-mediated photocatalytic oxidation of pollutants in oxygen-contained mediums has no selectivity due to the formation of reactive species on the irradiated surface of photocatalysts. As a result, almost all organic compounds can be photocatalytically oxidized. We selected acetone as a volatile organic compound for testing the synthesized materials because it does not cause itself the deactivation of photocatalysts and, consequently, is suitable for long-term experiments to evaluate their stability in the view of photocatalytic ability. In this case, a change in the rate of photocatalytic oxidation, that would be observed during experiments, can be reliably attributed to a change in photocatalyst itself due to its transformation.

Bismuth tungstate prepared via the hydrothermal method from bismuth nitrate and sodium tungstate was able to provide the oxidation of acetone vapor both under UV (371 nm) and visible light (450 nm). Carbon dioxide was the major oxidation product but small amounts of intermediate products, namely formaldehyde and formic acid, were also detected in the gas phase during experiments using IR spectroscopy. These products of incomplete oxidation have low threshold limits and are harmful for human health [12]. Formation of intermediates during the oxidation of pollutants over Bi_2_WO_6_ is a drawback for its application as a single-phase photocatalyst. In contrast to Bi_2_WO_6_, N-doped TiO_2_ prepared from titanium oxysulfate using ammonia as a nitrogen source resulted in the complete oxidation of acetone without the formation of gaseous intermediates under the radiation in both spectral regions. The photocatalytic activity of TiO_2_-N under UV attributes to the excitation of electrons from VB to the CB of anatase TiO_2_. Under visible light, the excitation occurs using intermediate energy levels of nitrogen species in the TiO_2_ lattice that are located higher than the VB of TiO_2_. Therefore, a lower energy of photons is required for the excitation of electrons compared to the band-to-band excitation. As mentioned above, the redox potentials of charge carriers photogenerated under visible light remain high enough to provide the complete oxidation of pollutants with a high rate. Considering single materials, TiO_2_-N is more active than Bi_2_WO_6_ in both spectral regions: in four times under UV and in 1.6 times under blue light (Figure 7). A high surface area of TiO_2_-N and a high amount of light absorbed in the visible region give positive effects on its activity and ability for complete oxidation. It should be noted that one of the best commercially available photocatalysts for visible light, namely KRONOClean® 7000 from Kronos Worldwide Inc. (Dallas, TX, USA), exhibited the visible-light activity of 0.07 μmol min^−1^ under the same conditions that is much lower than the corresponding values for Bi_2_WO_6_ (0.45 μmol min^−1^) and TiO_2_-N (0.70 μmol min^−1^). This fact confirms the high level of activity achieved for the synthesized materials.

An enhanced activity was achieved for the synthesized Bi_2_WO_6_–TiO_2_-N composites compared to the parent single-phase materials. It corresponds to the previously published results on an enhanced activity of Bi_2_WO_6_–TiO_2_ composites in the degradation of organic dyes, 2-nitrophenol, and acetaldehyde [51,52]. Similar trends both under UV and visible light were observed for the synthesized Bi_2_WO_6_–TiO_2_-N samples. The photocatalytic activity increased as the content of Bi_2_WO_6_ in composites was increased, but at a further increase in the Bi_2_WO_6_ content, activity began to decrease until the level for single Bi_2_WO_6_. Figure 7 clearly shows this dome-shaped dependance with a maximum at the molar ratio of 5:100 between the components for both spectral regions. The UV-light activity of this composite was higher than the activities of single TiO_2_-N and Bi_2_WO_6_ in 1.6 and six times, respectively. Under visible light, initial activity was increased in 1.4 and 2.2 times compared to TiO_2_-N and Bi_2_WO_6_, respectively.

An enhanced activity of Bi_2_WO_6_–TiO_2_-N composites results from the efficient separation of photogenerated charge carriers due to a heterojunction between the semiconductors. According to the estimations using an empirical equation based on the electronegativity and band gap of semiconductors, the energy bands of Bi_2_WO_6_ are located lower than the bands of TiO_2_. As a result, the heterojunction of type II, when the electrons photogenerated in the CB of TiO_2_ are transferred to the CB of Bi_2_WO_6_, while the holes migrate backwards from the VB of Bi_2_WO_6_ to the VB of TiO_2_, is commonly proposed for the Bi_2_WO_6_–TiO_2_ system [41,53]. An increase in the content of Bi_2_WO_6_ leads to an increase in the number of contacts between Bi_2_WO_6_ and TiO_2_-N nanoparticles, which has a positive effect on the photocatalytic activity. At a high content, Bi_2_WO_6_ begins to prevail and the system goes to its characteristics that are substantially lower compared to TiO_2_-N. Therefore, the Bi_2_WO_6_–TiO_2_-N composites with a low content of Bi_2_WO_6_ exhibit the highest activity.

In addition to a high activity, the stability of photocatalyst under long-term irradiation is also important for its practical application. N-doped TiO_2_ containing nitrogen impurities in the interstitial positions of the TiO_2_ lattice has a drawback in this aspect because a gradual decrease in its activity occurs under highly powerful radiation. Figure 8 illustrates this behavior of TiO_2_-N under visible light. The photocatalytic activity of pristine TiO_2_-N decreased by 47% compared to the initial level after 16 h of irradiation with blue light having a specific radiation power of 160 mW cm^−2^. This deactivation results from a partial oxidation of nitrogen species in the TiO_2_ lattice by the photogenerated holes that do not migrate to the surface of the photocatalyst and do not participate in the redox reactions with adsorbed molecules [25]. It is important to note that after longer irradiation for 2–3 days, the activity of TiO_2_-N reached a permanent level, namely 50% of the initial value, and did not change over further time (see Appendix A).

In contrast to TiO_2_-N, the synthesized Bi_2_WO_6_ exhibited a high stability from the first moment because no decrease in the values of the oxidation rate was observed during the irradiation under the same conditions (Figure 8a). Stability of Bi_2_WO_6_ arises from the fact that, in contrast to TiO_2_-N, it has a narrower band gap and the absorption of blue light corresponds to the band-to-band excitation of electrons. No substantial change in the chemical composition of semiconductors occurs in this situation. Similar behavior was achieved for the Bi_2_WO_6_–TiO_2_-N composites. The visible-light activity of composites with the Bi_2_WO_6_:TiO_2_-N molar ratio of 5:100 and higher did not decrease during long-term irradiation but even slightly increased similarly to Bi_2_WO_6_ (Figure 8a).

Deactivation of TiO_2_-N results from the oxidation of nitrogen species in the TiO_2_ lattice by the photogenerated holes. Therefore, the fast transfer of holes from the TiO_2_ phase may suppress the degradation of nitrogen species and, consequently, deactivation of the photocatalyst. An enhanced transfer of photogenerated holes can be realized in heterojunction systems when the position of VB in the second semiconductor is higher than the level of holes in TiO_2_ (type II) or when the holes recombine with the photogenerated electrons of the second semiconductor (Z-scheme). As mentioned above, the heterojunction of type II is commonly proposed for the Bi_2_WO_6_–TiO_2_ system (Figure 9a) but the energy bands of Bi_2_WO_6_ are located lower than the bands of TiO_2_, which prevents the transfer of holes from TiO_2_. At the same time, the energy levels of nitrogen species in TiO_2_-N are located substantially higher than the VB of TiO_2_ [54] and a Z-scheme heterojunction, when the photogenerated electrons from the CB of Bi_2_WO_6_ would interact with the photogenerated holes from nitrogen levels of TiO_2_ (Figure 9b), can be proposed for Bi_2_WO_6_–TiO_2_-N similarly to other systems [55,56,57]. Therefore, this junction can promote a removal of photogenerated holes from the nitrogen energy levels and suppress the degradation of nitrogen impurities by these holes.

In contrast to the samples mentioned above, the visible-light activity of the Bi_2_WO_6_–TiO_2_-N composite with a lower content of Bi_2_WO_6_ (i.e., molar ratio of 1:100) decreased during long-term irradiation, but this effect was much weaker than for pristine TiO_2_-N. For a detailed comparison, Figure 8b shows the relative values of activity, which were recalculated by normalization to the initial value of activity for each photocatalyst. After the first 9 h of irradiation, the activity of the Bi_2_WO_6_–TiO_2_-N (1:100) sample decreased by 17% compared to the initial value, but no substantial change was observed after further irradiation. No crystal phase of Bi_2_WO_6_ in the samples with low Bi_2_WO_6_ content can be a reason for the lower stability of these samples. This once again underlines the importance of the efficient heterojunction of charge carries between the semiconducting components in the composite system. Considering the data on the activity and stability for composite photocatalysts, as shown in Figure 7 and 8, a molar content of ca. 5% can be selected as an optimum content of Bi_2_WO_6_ in this system because at this value, a high activity and a high stability are achieved simultaneously.

The results described above show that the Bi_2_WO_6_–TiO_2_-N composite system exhibits an enhanced activity and stability compared to TiO_2_-N photocatalysts alone and has the potential for application in air purification using the photocatalytic oxidation method.

## 4. Conclusions

N-doped TiO_2_ (anatase, 98 m^2^ g^−1^) prepared from titanium oxysulfate using ammonia as a nitrogen source and bismuth tungstate (Bi_2_WO_6_, 34 m^2^ g^−1^) prepared via the hydrothermal method from bismuth nitrate and sodium tungstate are able to provide the complete oxidation of volatile organic compounds in air both under UV and visible light. N-doped TiO_2_ is more active than Bi_2_WO_6_ in both regions. A combination of TiO_2_-N and Bi_2_WO_6_ results in a synergistic effect because the composite with a low content of Bi_2_WO_6_ (5 mol.%) exhibits the photocatalytic activity that is substantially higher than activity of each component. A dome-shaped dependance of activity on the content of Bi_2_WO_6_ was found and the Bi_2_WO_6_–TiO_2_-N composite with a molar ratio of 5:100 had the highest activity among all the studied samples both under UV and visible light. In addition to an enhanced activity, Bi_2_WO_6_ in composites promotes N-doped TiO_2_ and prevents the degradation of nitrogen impurities in the TiO_2_ lattice by holes during the photocatalytic process. As a result, Bi_2_WO_6_–TiO_2_-N composites exhibit stable values of photocatalytic activity under highly powerful radiation (160 mW cm^−2^), while the activity of N-doped TiO_2_ is decreased by two times in 16 h under the same conditions. The enhanced activity and stability of Bi_2_WO_6_–TiO_2_-N composites result from an efficient separation of photogenerated charge carriers due to a heterojunction between the semiconductors. Therefore, the Bi_2_WO_6_–TiO_2_-N composite system has the potential for efficient purification of air using the photocatalytic oxidation method.

## Figures and Tables

**Figure 1 nanomaterials-12-00359-f001:**
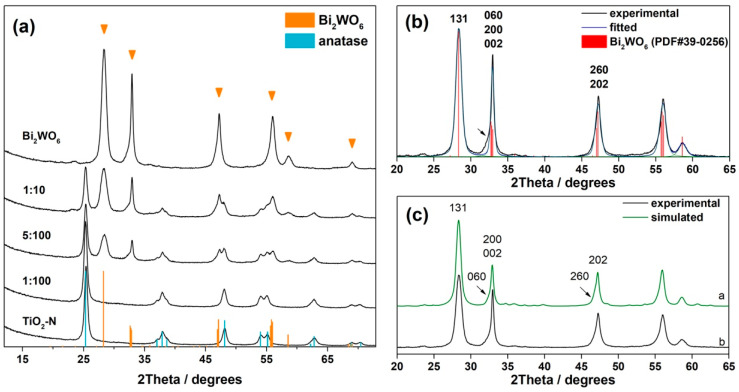
XRD patterns of the synthesized TiO_2_-N, Bi_2_WO_6_, and composite photocatalysts (**a**). Analysis (**b**) and simulation (**c**) of XRD pattern of Bi_2_WO_6_ sample.

**Figure 2 nanomaterials-12-00359-f002:**
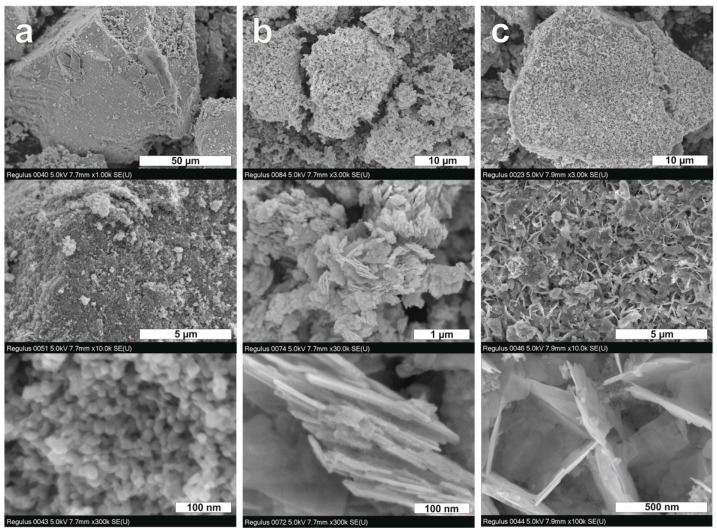
SEM micrographs of TiO_2_-N (**a**), Bi_2_WO_6_ (**b**), and Bi_2_WO_6_–TiO_2_-N (5:100) composites (**c**) at different magnifications.

**Figure 3 nanomaterials-12-00359-f003:**
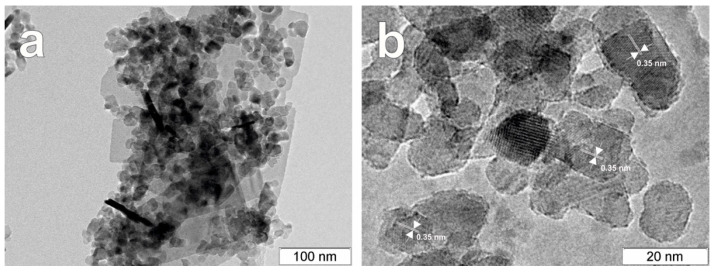
TEM images of Bi_2_WO_6_–TiO_2_-N (5:100) sample at different magnifications (**a**,**b**).

**Figure 4 nanomaterials-12-00359-f004:**
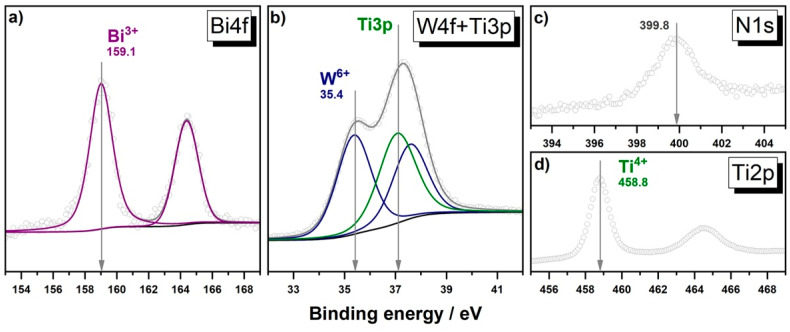
Photoelectron Bi4d (**a**), W4f (**b**), N1s (**c**), and Ti2p (**d**) spectral regions for the sample of Bi_2_WO_6_–TiO_2_-N (5:100).

**Figure 5 nanomaterials-12-00359-f005:**
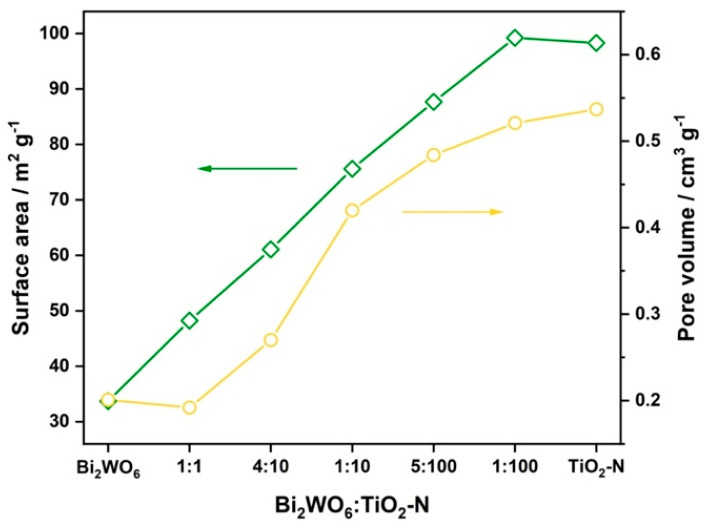
Effect of Bi_2_WO_6_:TiO_2_-N molar ratio on the textural characteristics of materials.

**Figure 6 nanomaterials-12-00359-f006:**
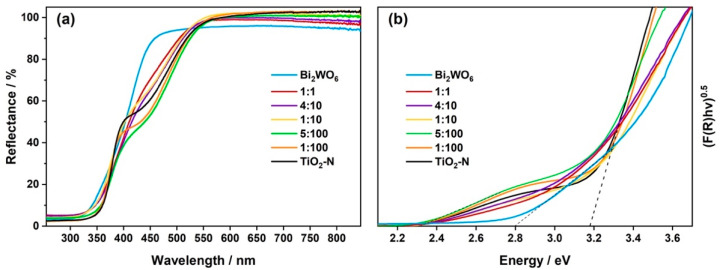
UV-vis DRS spectra of the synthesized materials (**a**) and corresponding Tauc plots (**b**).

**Figure 7 nanomaterials-12-00359-f007:**
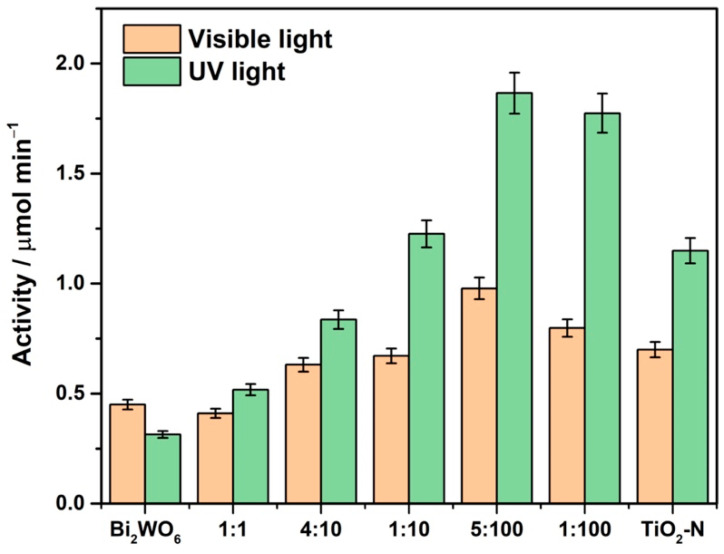
Effect of Bi_2_WO_6_:TiO_2_-N ratio on the photocatalytic activity of composites under UV and visible light.

**Figure 8 nanomaterials-12-00359-f008:**
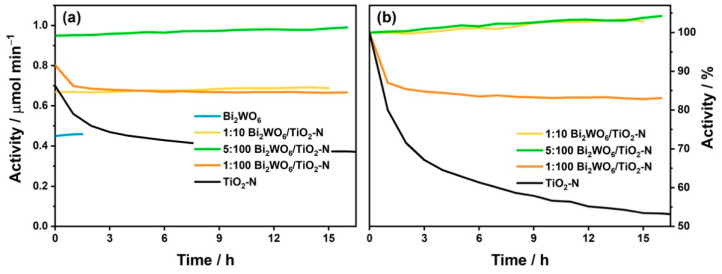
**Figure 8**. Effect of irradiation time on visible-light activity of the prepared materials in absolute (**a**) and relative (**b**) units.

**Figure 9 nanomaterials-12-00359-f009:**
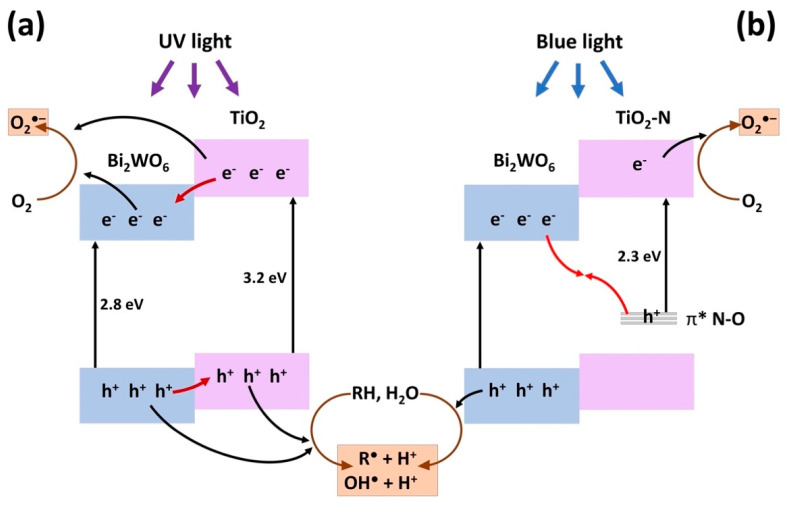
Proposed pathways of photogenerated charge carriers in Bi_2_WO_6_–TiO_2_ (**a**) and Bi_2_WO_6_–TiO_2_-N (**b**) systems.

**Table 1 nanomaterials-12-00359-t001:** Content of precursors during the synthesis of photocatalysts.

Bi_2_WO_6_:TiO_2_-N (Molar Ratio)	m(X), g (Bi(NO_3_)_3_·5H_2_O)	m(Y), (Na_2_WO_4_·2H_2_O)	m(Z), g (TiO_2_-N)
1:0 (Bi_2_WO_6_)	2.42 (2.5 mmol)	0.82 (1.25 mmol)	-
1:1	3.03 (6.25 mmol)	1.03 (3.125 mmol)	0.25 (3.125 mmol)
4:10	1.21 (2.5 mmol)	0.41 (1.25 mmol)	0.25 (3.125 mmol)
1:10	1.21 (2.5 mmol)	0.41 (1.25 mmol)	1 (12.5 mmol)
5:100	0.6 (1.25 mmol)	0.2 (0.625 mmol)	1 (12.5 mmol)
1:100	0.12 (0.25 mmol)	0.04 (0.125 mmol)	1 (12.5 mmol)

## Data Availability

The data presented in this study are available upon request from the corresponding author. The data are not publicly available due to privacy.

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
