# Peer review of "Enhanced Photocatalytic Activity and Stability of Bi2WO6 – TiO2-N Nanocomposites in the Oxidation of Volatile Pollutants"

_nanomaterials, 2022, doi:10.3390/nano12030359_

Round 1
Reviewer 1 Report
The authors submitted an interesting manuscript on the enhanced photocatalytic activity and stability of Bi2WO6 – TiO2-N nanocomposites in oxidation of volatile pollutants. As a whole the paper is well organized and the results are sound and well supported by experimental evidence. I recommend publication after some minor amendments:
1. The introduction should be much more focused on the problem statement and motivation along with the proposed solutions;
2. The discussion of results lacks a thorough analysis, it is mostly a description of what can be inferred from the plots.
Author Response
We would like to thank the reviewer for valuable comments.
- We added information and improved discussion in the introduction section to be more focused on the concerned problem and motivation for study.
- We improved the discussion of experimental results and comparison with literature data.
Reviewer 2 Report
This manuscript reports the synthesis and characterization of Bi2WO6-TiO2-N composites and their photocatalytic activity in oxidation of acetone, a volatile pollutant. The authors show that the combination of N-doped TiO2 and Bi2WO6 affords a very stable composite photocatalyst. The research was conducted appropriately, the authors provide a detailed structural analysis and discussion of the photocatalytic properties of the new materials. The results are presented in clear form and supported by the experiments and thus the manuscript is appropriate to be published in Nanomaterials.
I suppose that the authors have planned new experiments with this type of nanocomposites to test their photocatalytic properties against other volatile compounds beyond acetone, so that they can verify their activity against other pollutants at higher temperatures. In that case, I would recommend synthesizing new materials (and studying their photocatalytic properties) with Bi2WO6 percentages between 1% and 5%, since this range is where the maximum activity is probably found both under visible and ultraviolet light.
Author Response
We would like to thank the reviewer for this valuable suggestion. We will follow it in future studies.
Reviewer 3 Report
The manuscript “Enhanced Photocatalytic Activity and Stability of Bi2WO6 – TiO2-N Nanocomposites in Oxidation of Volatile Pollutants is well written and organized and of high interest to the readers. However, the discussion of the results with existing literature should be enhanced.
Authors have chosen Acetone as a volatile organic compound for testing the synthesized photocatalysts, because it does not cause deactivation of the catalysts over long-term experiments. How about the performance and stability for organic pollutants? Can it be generalized for other organic molecules? Based on what? Please explain.
The stability of the proposed catalyst was studied over 16 h. This is a short time. Longer periods should be tested, days, months. How about regenerating the catalysts? And what is the destiny of these catalysts after losing the activity?
Also toxicological evaluation would be interesting.
Author Response
We would like to thank the reviewer for valuable comments.
- We improved the discussion of experimental results and comparison with literature data.
- Along with isopropyl alcohol, diethyl ether, hexane etc,. acetone is commonly used as organic solvent in industry and is one of the volatile organic pollutants. There are some peculiarities in the TiO2-mediated photocatalytic oxidation of different types of pollutants [D.S. Selishchev, N.S. Kolobov, A.A. Pershin, D.V. Kozlov, Appl. Catal. B Environ. 200 (2017) 503–513, M.N. Lyulyukin, P.A. Kolinko, D.S. Selishchev, D.V. Kozlov, Appl. Catal. B Environ. 220 (2018) 386–396.], but this method has no selectivity due to the formation of reactive species. As a result, almost all organic compounds can be oxidized on the irradiated surface of photocatalyst. Therefore, considering the mentioned peculiarities, the results of this study can be generalized for other volatile organic compounds.
- During the photocatalytic oxidation of acetone under UV light, TiO2 is stable for years. This was a reason for its selection as a test oxidizing compound. For investigation of TiO2-N under visible light, the experiments on stability were performed under blue light of a high radiation power (160 mW cm-2), which is 1.6 times more powerful than incident solar light (100 mW cm-2), for more visible effect and faster achievement of a steady state. A decrease in the activity of pristine N-doped TiO2 occurs in first 15 hours. Under longer irradiation, its activity reaches a permanent level, 50% of initial value, and does not change further (see Figure S4 in Supplementary). The regeneration of TiO2-N is not possible because its deactivation is related to the internal irreversible transformations. A decrease in the activity of Bi2WO6/N-doped TiO2 (1:100) occurs in first 6 hours, but further its activity does not change. Activity of composites with ratios more than 1:100, as well as pristine Bi2WO6 does not decrease from the first moment of irradiation. For all photocatalysts, a change from initial activity to steady-state activity is shown that is the key point because after reaching steady state conditions these photocatalysts can work for a long time. Therefore, the studied period of time in stability tests is relevant for the purpose of this study.
- N-doped TiO2 similarly to pristine TiO2, as well as Bi2O3 and WO3 are non-toxic, but irritant if inhale and swallowed:
https://www.sigmaaldrich.com/RU/en/sds/aldrich/232033
https://www.sigmaaldrich.com/RU/en/sds/sial/95381
https://www.sigmaaldrich.com/RU/en/sds/aldrich/204781
Concerning bismuth tungstate, to the best of our knowledge, there is no information on its toxicity.
Reviewer 4 Report
The work described here is interesting and fits with the scopre of the journal. It can be published after minor revision.
In the comparison of the photocatalytic activity of Figure 7 it is not clear if there is a statistical difference between the different samples, some samples do not seem statistically different to me - this point should be clarified and some statistical analysis added.
Minor points: in Figure 1a the symbols indicating the phases should be larger. Moreover in the SEM images in Figure 2 the scale bars should be made more visible.
Author Response
We would like to thank the reviewer for valuable recommendations.
- Based on the statistics of many experiments, a relative error estimated using confidence interval for the probability of 95% during the measurement of photocatalytic activity in the described setup did not exceed 5%. This error value was used for all samples to evaluate statistically significant difference in their activity. Information on the statistics in measurement of activity was added to the text of manuscript. Additionally, error bars in Figure 7 were made thicker for better visibility.
- Figure 1a and 2 were improved according to the recommendations of reviewer.
Reviewer 5 Report
The manuscript was prepared according to the rules. It concerns the testing of the stability of the composite material under the flow conditions. The following remarks are for clarification:
- the presented composite material cannot be called heterojunction because both TiO2 and Bi2WO6 are n-type semiconductors. Heterojunction is in the case of a p-n junction
- stability should be tested by providing a new portion of the contaminant, and so on, for example, in four long tests
- photocatalytic processes should be tested using a different type of pollutant, e.g. phenol
- the actual ratio between TiO2 and Bi2WO6 should be given eg from the XPS results
Author Response
We would like to thank the reviewer for valuable comments. Please find the responses below:
- We agree that classical heterostructure corresponds to a layered structure consisted of p- and n-type semiconductors, and heterojunction is charge transfer (junction) between these semiconductors, which separately have different positions of energy bands. But currently, especially in the field of photocatalysis, the term “heterojunction” is commonly used for both p-n and non p-n semiconducting systems. As a confirmation of this statement, we provide a list of references to top-cited or specialized papers concerning this question:
- Wang, H.; Zhang, L.; Chen, Z.; Hu, J.; Li, S.; Wang, Z.; Liu, J.; Wang, X. Semiconductor Heterojunction Photocatalysts: Design, Construction, and Photocatalytic Performances. Chemical Society Reviews 2014, 43, 5234–5244, doi:10.1039/C4CS00126E.
- Yuan, Y.P.; Ruan, L.W.; Barber, J.; Joachim Loo, S.C.; Xue, C. Hetero-Nanostructured Suspended Photocatalysts for Solar-to-Fuel Conversion. Energy & Environmental Science 2014, 7, 3934–3951, doi:10.1039/C4EE02914C.
- Xu, Q.; Zhang, L.; Yu, J.; Wageh, S.; Al-Ghamdi, A.A.; Jaroniec, M. Direct Z-Scheme Photocatalysts: Principles, Synthesis, and Applications. Materials Today 2018, 21, 1042–1063.
- Xu, Q.; Zhang, L.; Cheng, B.; Fan, J.; Yu, J. S-Scheme Heterojunction Photoctalyst. Chem 2020, 6, 1543–1559, doi:10.1016/J.CHEMPR.2020.06.010.
- Low, J.; Yu, J.; Jaroniec, M.; Wageh, S.; Al-Ghamdi, A.A. Heterojunction Photocatalysts. Advanced Materials 2017, 29, 1601694, doi:10.1002/ADMA.201601694.
- Marschall, R. Semiconductor Composites: Strategies for Enhancing Charge Carrier Separation to Improve Photocatalytic Activity. Advanced Functional Materials 2014, 24, 2421–2440, doi:10.1002/ADFM.201303214.
- Francesca Pinto, Anna Wilson, Benjamin Moss, and Andreas Kafizas, Systematic Exploration of WO3/TiO2 Heterojunction Phase Space for Applications in Photoelectrochemical Water Splitting/ The Journal of Physical Chemistry C Article ASAP, DOI: 10.1021/acs.jpcc.1c08403
In the whole manuscript, we do not use the term “heterostructure” but we use the term “heterojunction” to indicate the interface charge transfer between semiconducting components of composite photocatalyst.
- The mentioned suggestion can be mainly attributed to the photocatalytic experiments into a batch reactor where the concentration of pollutant is monotonically decreased under irradiation, and we have to use initial values of photocatalytic rate for valid comparison of activity. In our case, the photocatalytic experiments were performed using a continuous-flow setup where new portion of oxidizing compound is injected into the photoreactor every moment with inlet flow. In this case, the photocatalyst works under steady-state conditions, and there is no effect of pollutant’s concentration on the reaction rate. As a result, we can check the stability of photocatalyst by evaluation of reaction rate for long time.
- This study is focused on the design of materials to produce highly active and stable photocatalysts for the oxidation of gas-phase pollutants. TiO2-mediated photocatalytic oxidation of contaminants in oxygen-contained mediums has no selectivity, and almost all organic compounds can be oxidized over TiO2-contained photocatalysts. Some peculiarities would be observed for different types of pollutants. Therefore, we selected certain compound and performed the photocatalytic experiments under optimal conditions when the operation parameters (e.g., mass of photocatalyst, concentration of oxidizing compound) had low influence on the rate of reaction. It allowed us to compare different materials correctly for valid analysis of correlations between the physicochemical characteristics and the activity of materials and selection of best ones. For this reason, testing other pollutants (especially, water pollutant) is beyond the scope of this paper and will be the subject of our future studies.
- The photocatalysts were prepared via the addition of TiO2-N sample of required mass to the solution of Bi(NO3)3followed by mixing with solution Na2WO6, which both. Bi2WO6 is sparingly soluble material. As a result, it precipitates almost quantitatively after mixing solutions of precursors, Bi(NO3)3 and Na2WO6. Therefore, content of TiO2-N and Bi2WO6 components in the prepared materials was close to the values estimated from loadings. It was confirmed by X-ray fluorescence analysis. The mentioned information was added to the text of revised manuscript.
Round 2
Reviewer 3 Report
The manuscript was improved and I agree with its publication.